# Comparative Metabolomics Analysis of Stigmas and Petals in Chinese Saffron (*Crocus sativus*) by Widely Targeted Metabolomics

**DOI:** 10.3390/plants11182427

**Published:** 2022-09-17

**Authors:** Lin Zhou, Youming Cai, Liuyan Yang, Zhongwei Zou, Jiao Zhu, Yongchun Zhang

**Affiliations:** 1Shanghai Key Laboratory of Protected Horticulthural Technology, Forestry and Pomology Research Institute, Shanghai Academy of Agricultural Sciences, Shanghai 201106, China; 2Department of Biology, Wilfrid Laurier University, Waterloo, ON N2L 3C5, Canada

**Keywords:** *Crocus sativus*, stigmas, petals, metabolomics analysis

## Abstract

The dried stigmas of *Crocus sativus*, commonly known as saffron, are consumed largely worldwide because it is highly valuable in foods and has biological activities beneficial for health. Saffron has important economic and medicinal value, and thus, its planting area and global production are increasing. Petals, which are a by-product of the stigmas, have not been fully utilized at present. We compared the metabolites between the stigmas and petals of *C. sativus* using a non-targeted metabolomics method. In total, over 800 metabolites were detected and categorized into 35 classes, including alkaloids, flavonoids, amino acids and derivatives, phenols and phenol esters, phenylpropanoids, fatty acyls, steroids and steroid derivatives, vitamins, and other metabolites. The metabolite composition in the petals and stigmas was basically similar. The results of the study showed that the petals contained flavonoids, alkaloids, coumarins, and other medicinal components, as well as amino acids, carbohydrates, vitamins, and other nutritional components. A principal components analysis (PCA) and an orthogonal partial least-squares discriminant analysis (OPLS-DA) were performed to screen the different metabolic components. A total of 339 differential metabolites were identified, with 55 metabolites up-regulated and 284 down-regulated. The up-regulated metabolites, including rutin, delphinidin-3-*O*-glucoside, isoquercitrin, syringaresinol-di-*O*-glucoside, dihydrorobinetin, quercetin, and gallocatechin, were detected in the petals. The down-regulated metabolites were mainly glucofrangulin B, acetovanillone, daidzein, guaiazulene, hypaphorine, indolin-2-one, and pseudouridine. KEGG annotation and enrichment analyses of the differential metabolites revealed that flavonoid biosynthesis, amino acids biosynthesis, and arginine and proline metabolism were the main differentially regulated pathways. In conclusion, the petals of *C. sativus* are valuable for medicine and foods and have potential utility in multiple areas such as the natural spice, cosmetic, health drink, and natural health product industries.

## 1. Introduction

*Crocus sativus* L., commonly known as saffron or Zang-Hong-Hua, is a perennial herb of the genus *Crocus* in the Iridaceae family, which is widely cultivated in many countries around the Mediterranean Sea and parts of Asia [1]. The dried stigma of *C. sativus*, called saffron, is widely used for cooking as a dietary spice and as a food colorant because of several important bioactive constituents including crocin (the color of saffron), safranal (the odor of saffron), and picrocrocin (the taste of saffron) [2,3]. Although China began to plant *C. sativus* in the 1980s [4], saffron had been used in traditional herbal medicine for a long time. Recently, saffron has been widely studied for its valuable medical properties. An increasing number of studies indicated the potential uses of saffron extract as an adjuvant in anti-cancer, anti-diabetic, anti-depressant, and anti-inflammatory treatments [3,5]. With great interest in and concern for its health benefits, there is an increasing need for saffron. Thus, the plantation areas of *C. sativus* have expanded unceasingly worldwide, and saffron yields have increased year by year in order to meet the market demand [6,7].

Although saffron is composed of multiple components, the previous studies have focused only on crocetin and its glucosidic derivatives, little in-depth research has been conducted on the efficacy of other components. Additionally, the petals of *C. sativus* have a distinctive rich, unique aroma, but few studies have reported on the functional ingredients of petals. Despite the fact that petals are harvested in large quantities during the flowering period, there is no recent industrial application [8] for them. However, an increasing number of studies confirmed that the aqueous extract of saffron petals had great potential in the prevention of cardiovascular diseases [3], as an antioxidant [9], and in ameliorating the symptoms of polycystic ovary syndrome [10]. Plants have a diverse array of metabolic products generated from both basic substances and products of secondary metabolism, such as carbohydrates, proteins, peptides, lipids, minerals, vitamins, amino acids, and organic acids. In recent years, in order to fully develop and apply petals, researchers have begun to focus on the composition of petals. By using high-performance liquid chromatography-diode array detection-mass spectrometry (HPLC-DAD), ultra-performance liquid chromatography-quadrupole time-of-flight mass spectrometry (UPLC-QTof-MS/MS), and other methods, the results of multiple studies indicate that the petals of *C. sativus* are rich in phenolic, flavonoids, and anthocyanin [11,12]. Quite a few scholars have carried out research on the composition and content of flavonoid, polyphenols, anthocyanin, and other components and on the optimization of extraction methods [13,14]. However, these methods can only measure known, single, or small amounts of metabolites [15,16]. To date, a comprehensive analysis of the composition and metabolism of the stigmas and petals in *C. sativus* has not been conducted.

Traditional metabolomics analysis is an effective technology used for the determination of metabolites and can be divided into untargeted and targeted. In recent years, metabolomics as a well-established method to characterize plant metabolism has been increasingly applied in the study of Chinese medicinal herbs [17], such as *Epimedium brevicornum* [18], *Echinacea* plants [19], *Curcumae* Radix [20], *Alisma orientale* [21], *Taxus* species [22], and *Pueraria lobata* [23]. For example, the metabolite composition of and differences between *Pueraria lobata* and its two varieties were determined by metabolomics analysis, providing a better understanding of the nutritional and medicinal variations in *P. lobata* [23]. As modern technology advances rapidly, widely targeted metabolomic analysis, a novel approach that combines the advantages of non-targeted metabolomics and targeted metabolomics, has made possible a prompt and ultra-sensitive detection of a huge number of metabolites. To the best of our knowledge, no relevant metabolomics research on *C. sativus* has been reported thus far.

In this study, widely targeted metabolomics was employed to profile the composition of metabolites in the stigmas and petals. The results of our study will provide insights into the formation mechanisms of functional ingredients and lay a theoretical foundation for the extraction of functional components and for product development.

## 2. Results

### 2.1. Widely Targeted Metabolomics Analysis in Stigmas and Petals

The stigmas and petals of the *C. sativus* samples were collected and used for metabolomics analysis. The sampling stages and plant materials are the same as described in Figure 1. Over 800 metabolites were detected in the stigmas and petals and categorized into 35 broad categories (Appendix A) based on a self-compiled database (Shanghai Biotree Biotech Co., Ltd., Shanghai, China), and the PubChem and HMDB databases, including alkaloids, amino acids and derivatives, chalcones, coumarins, flavonoids, organic acids, phenols and phenol esters, vitamins, etc. (Figure 2).

In total, 827 metabolites were identified from the stigmas, including 92 alkaloids; 42 amino acids and derivatives; 7 anthraquinones; 16 benzene and substituted derivatives; 7 carbohydrates; 16 carboxylic acids and derivatives; 6 chalcones; 3 cholines; 5 cinnamic acids and derivatives; 26 coumarins; 18 diterpenoids; 34 fatty acyls; 146 flavonoids (including flavone, flavanone, and flavonoid); 5 indoles and derivatives; 8 iridoids; 4 keto acids and derivatives; 12 lignans; 5 lipids; 14 monoterpenoids; 26 nucleotide and its derivates; 11 organic acids; 7 organonitrogen compounds; 27 organooxygen compounds; 70 phenols and phenol esters; 40 phenylpropanoids; 8 phytohormone; 8 prenol lipids; 23 pteridines and derivatives, purine nucleosides, pyridines and derivatives, pyrimidine nucleosides, pyrimidine nucleotides, and pyrrolopyrimidines; 12 quinones; 20 sesquiterpenoids; 38 steroids and steroid derivatives; 21 triterpenoids; 6 vitamins; 11 xanthones; and 33 other metabolites.

There are 824 metabolites in the petals. Among them, 821 are mostly consistent with those in the stigmas. Of the 827 metabolites in the stigmas, 6 metabolites ((+)-affinisine, xanthyletin, nervonic acid, pseudouridine, hygromycin B, and leucodin) were unique to the petals. Among the petals, three metabolites (quassin, rutin, and syringaresinol-di-*O*-glucoside) were different from those of the stigmas.

### 2.2. Multivariate Analysis of Differential Metabolites among Stigmas and Petals

The contribution ratio of principal component 1 (PC1) was 63.9%, and that of PC2 was 7.5%. The samples were tightly clustered together, indicating a high repeatability of the experimental methods. In this PCA, the petal samples were clearly separated from the stigma samples, indicating significant differences between the tissues (Figure 3a).

To further identify the differences in the composition of the metabolites, the OPLS-DA model was employed to characterize the differential metabolites between the stigmas and petals. In the OPLS-DA model, the values of R^2^X, R^2^Y, and Q^2^ of the model were 0.755, 1, and 0.988, respectively. From the OPLS-DA score scatter plot (Figure 3b), the stigmas and petals were located on the positive and negative sides of the first principal component, indicating that the samples were significantly different.

Compared with the stigmas, 339 differentially expressed metabolites were identified in the petals, of which 55 metabolites were up-regulated and 284 were down-regulated. A volcano plot was employed to visualize the differences in metabolite levels (Figure 4). The assignment of differential metabolites was identified based on the published literature and databases such as the HMDB and PubChem compound databases. The up-regulated metabolites in the petals included 23 flavonoids (including flavone, flavanone, and flavonoid); 5 alkaloids; 5 steroids and steroid derivatives; 3 amino acids and derivatives; 3 fatty acyls; 3 diterpenoids; and 13 others, including rutin, delphinidin-3-*O*-glucoside, isoquercitrin, syringaresinol-di-*O*-glucoside, dihydrorobinetin, quercetin, and gallocatechin (Figure 5, Appendix A). The top 55 down-regulated metabolites in the petals included mainly 9 flavonoids; 7 alkaloids; 5 coumarins; 5 sesquiterpenoids; 3 organooxygen compounds; 3 phenols; 3 phenylpropanoids; 2 carbohydrates; 1 benzene and substituted derivatives; 1 amino acid and derivatives; and 16 others, including glucofrangulin B, acetovanillone, daidzein, guaiazulene, hypaphorine, indolin-2-one, and pseudouridine (Figure 6, Appendix A).

### 2.3. Metabolic Pathway Analysis of Differential Metabolites

A total of 339 differentially expressed metabolites were annotated in 61 biological pathways in the KEGG database (Appendix A). These KEGG pathways are mainly involved in carbon metabolism; biosynthesis of amino acids; biosynthesis of secondary metabolites; and other metabolic processes (such as flavonoid biosynthesis; amino acids biosynthesis; arginine and proline metabolism; phenylalanine, tyrosine, and tryptophan biosynthesis; glycolysis and gluconeogenesis; and amino sugar and nucleotide sugar metabolism).

The pathway enrichment analysis demonstrated that 52 metabolic pathways were enriched (Figure 7). The top five highly enriched pathways are beta-alanine metabolism; pantothenate and CoA biosynthesis; glycine, serine and threonine metabolism; flavone and flavonol biosynthesis; and flavonoid biosynthesis. Taken together, KEGG pathways are mostly enriched in flavonoid biosynthesis. In Figure 8 and Figure 9, quercetin in the petals was significantly up-regulated in flavone, flavonol, and flavonoid biosynthesis.

## 3. Discussion

The dried stigmas of *C. sativus* are widely used in the food, medical, and chemical industries. Both the cultivation region and the yield growth are increasing year by year to fulfill the market demand [7]. The petals of *C. sativus* are the main by-product of saffron processing, which is produced at a high level but is not applied and thrown out [24], mainly because of an insufficient understanding of metabolite components. However, recent studies pay much more attention to crocetin and its glucosidic derivatives in stigmas, without considering the whole metabolite composition. Metabolomics is one of the developing “-omics” technologies and has enabled the generation of large-scale metabolomics measures. In this study, we used the widely targeted metabolites based on the UPLC-MS/MS detection platform to analyze the metabolic differences between petals and stigmas. Although the present research only obtained a relative quantification of metabolite levels, we identified over 800 metabolites in the stigmas and petals. Thus, this study contributes to further understanding of the metabolite composition of stigmas and petals in the flowering stage of *C. sativus* and provides an experimental basis for the reasonable application of petals and stigmas.

At the flowering stage, both the stigmas and petals of *C. sativus* contain a variety of flavonoid compounds, such as apigenin, cianidanol, galangin, kaempferol, myricetin, quercetin, etc. Previous studies have shown that flavonoids are a versatile class of natural compounds and possess immense bioactive potential [25,26], while having nutraceutical, pharmaceutical, and cosmetic applications [27]. In addition, there are abundant alkaloids in stigmas and petals, such as galantamine, theobromine, and theophylline. Various coumarins were detected simultaneously, including bergapten, bergamotine, dalbergin, daphnoretin, isobergapten, neoglycyrol, notopterol, scoparone, and xanthyletin. Alkaloids and coumarins isolated from plants are commonly found to have a wide range of biological properties including anti-viral, anti-microbial, anti-oxidative, anti-inflammatory, and anti-cancer activities [28,29]. It can be seen that both the stigmas and petals are a good source of flavonoids, alkaloids, and coumarins with medicinal benefits, suggesting that the petals have the potential to be developed and utilized for their medicinal value.

As is well known, dried stigmas are widely used as dietary spices and food colorants [1,13]. Previous studies have suggested that saffron can be used as a dietary spice mainly because of its complex mixture of volatile and nonvolatile compounds and carotenoid derivatives [1]. In the present study, monoterpenoids and diterpenoids were detected in the stigmas and petals, which may also be important components in aroma composition. Terpenoids are a large class of naturally occurring compounds and have been widely used in the industrial or medicinal fields as flavorings, fragrances, and pharmaceuticals [30]. We detected more than 10 monoterpenoids, including picrocrocin, cineole, beta-citronellol, geraniol, and others. As an example, geraniol is an acyclic monoterpene alcohol that possesses a rose-like odor with a variety of applications such as flavoring or fragrance [31]. Diterpenoids, including sterebin A, sterebin E, rubusoside, and stevioside were also detected in this study. These natural sweet substances have been widely used as a natural sweetener for condiments and additives [32]. It can be seen that the petals have the potential to be used as spices or as raw materials in essential oils.

In recent years, saffron has been utilized as a common health drink due to its medicinal and health values. Our study found that, in addition to well-known components such as crocin, the stigmas and petals of *C. sativus* contain a number of water-soluble amino acids, carbohydrates, and vitamins, including isoleucine, glutathione oxidized, proline, d-Maltose, l-Gulose, d-Xylulose, vitamin A, and ascorbic acid. At present, the functional beverage industry has rapidly grown, becoming the biggest part of the functional food sector; a number of herbal-based functional beverages have been developed in the market to address various health problems [33]. *Dendranthema grandiflorum* [34], *Agrimonia asiatica* [33], *Aegle marmelos* [35], *Cyclopia subternata* [36], and other plant species have been developed into herbal beverages and have gained popularity among consumers who are concerned about their health. Saffron production is time-consuming and labor-intensive because it requires manual removal of stigmas from flowers. Processing 1 kg of saffron requires picking 200,000–400,000 stigmas, which takes 350–450 h [2]. Therefore, directly drying the whole flower for functional beverage development can not only reduce the pressure of stigma picking but also improve the utilization of petals. Currently, both the stigmas and petals of saffron have been used in the development of health drinks. A liquid beverage composed of ginseng, saffron, and acerola cherry extracts regulates qi and promotes blood circulation, improves digestive function, and relieves fatigue [37]. Saffron, jasmine, theanine, [γ]-aminobutyric acid, and wolfberry are used in the processing of healthy vegetable-based drinks that help with stress relief, improve quality of sleep, boost immune system and gastrointestinal motility, and regulate endocrine functions [38]. Flower tea made from a mixture of saffron petals, roses, and orange peel has significant preventive and therapeutic effects when treating depression [39]. As can be seen, functional beverages can be developed by using saffron and its components alone or in combination with ginseng, rose, goji berries, jasmine, chrysanthemum, tangerine peel, and licorice.

Although the stigmas and petals are similar in composition, there are differences in their metabolite contents. The up-regulated or down-regulated metabolites of the petals are mainly flavonoids, alkaloids, and coumarins. It is noteworthy that a total of 147 flavonoids metabolites were detected in the stigmas and petals, of which 23 were significantly up-regulated in petals, 9 were down-regulated, and 115 had no significant difference. Moreover, the relative amount of rutin, delphinidin-3-*O*-glucoside, dihydrorobinetin, quercetin, gallocatechin, and other flavonoids in the petals were five times greater than those in stigmas. Among them, quercetin and rutin are commercially used as dietary antioxidant supplements in food and beverages [40]. In this study, the KEGG pathway analysis revealed that flavone and flavonol biosynthesis, and the flavonoid biosynthesis signaling pathway were significantly enriched. Quercetin was involved in flavone, flavonol, and flavonoid biosynthesis; the up-regulation of quercetin in petals promotes the accumulation of myricetin, isoquercetin, and rutin. Petals contain high levels of anthocyanins, especially delphinidin-3-*O*-glucoside. Delphinidin-3-O-glucoside is an important metabolite in the formation of petal color. Alkaloids and coumarins showed similar results to those of the flavonoids, i.e., there was no significant difference in the content of most metabolites. Specifically, among the 92 alkaloids detected, the contents of the stigmas and petals were relatively similar, only 5 were significantly up-regulated in petals, and 7 were down-regulated. We detected a total of 26 coumarins in the stigmas and petals, of which xanthyletin was only detected in the stigmas, while isobergapten, decursinol, bergapten, and trioxsalen were down-regulated in the petals. Previous evidence found that the extracts of the petals exhibit versatile biological and pharmacological activities [16,41]. We speculate that it is closely related to the abundance of flavonoids, alkaloids, coumarins, and other metabolites in the petals. With further study and validation, the extract of petals can be used as a natural flavoring, coloring, and high-purity antioxidant, with possible uses in food industry, as well as for combating a number of human diseases or relieving ailments.

## 4. Materials and Methods

### 4.1. Site Description

The experiment was conducted at the Forestry and Pomology Research Institute, Shanghai Academy of Agricultural Sciences (31.22 N, 121.33 E), Shanghai, China from November 2020 to December 2021. The cultivation of *C. sativus* was carried out using a unique “two-segment” cultivation mode [4]. Specifically, from early November 2020 to late April 2021, corms of *C. sativus* were planted in a mixed matrix of 6:2:2 (*v*/*v*/*v*) peat, vermiculite, and perlite. An organic fertilizer was used as the field base fertilizer. The plants were fertilized every 20 days with 30:10:10 (NPK). In early May 2021, the corms of *C. sativus* were harvested when all leaves were wilted.

Five hundred corms weighting 25–35 g without visible damage were chosen for the experiment. From early May to the end of November 2021, the corms were placed in a plastic crate (42 cm × 42 cm × 5 cm, L × W × H) without soil substrate and kept in the incubation room under controlled temperature, humidity, and light conditions. From 16 May to 16 September, the corms were placed in a permanent dark environment. From 17 September until blooming, the corms were exposed in 12 h light/dark cycles at temperatures of 25 °C during light and 15 °C during darkness, with 65–80% humidity.

### 4.2. Sample Preparation and Metabolite Extraction

Fresh stigmas and petals for metabolomics were collected between late October and early November 2021. Three independent biological replicates were tested for each sample. All samples were freeze-dried for approximately 48 h and crushed using a mixer mill with a zirconia bead for 30 s at 60 Hz. A 50 mg amount of powder of an individual sample was accurately weighed and transferred into an Eppendorf tube. Subsequently, 700 μL of the extract solution (methanol/water = 3:1, precooled at −40 °C) containing an internal standard (2-chloro-dl-phenylalanine, 1 μg/mL) was added; the mixture was stirred for 30 s on a Vortex-shaker, homogenized for 4 min at 35 Hz, and sonicated for 5 min in an ice-water bath. Then, the samples were incubated overnight at 4 °C on a shaker. Afterwards, the extract was centrifuged at 12,000 rpm at 4 °C for 15 min. Immediately after, the supernatant was collected and filtered with a 0.22 μm microporous membrane. The filtered supernatants were then diluted 20 times with methanol/water mixture (*v*:*v* = 3:1, containing internal standard), and the quality control (QC) sample was prepared by pooling 20 μL aliquots from all samples. All supernatants were stored at −80 °C until the UHPLC-MS analysis.

### 4.3. UHPLC-MS Analysis

The UHPLC separation was carried out on an Exion LC AD™ System (AB SCIEX, Concord, Ontario, Canada). The metabolites were separated with an ACQUITY UPLC HSS T3 C18 column (Waters, 2.1 mm × 100 mm × 1.8 μm) at 40 °C. The mobile phase A was 0.1% formic acid in water, and the mobile phase B was acetonitrile. The gradient program was as follows: 0–0.5 min 98% A, 0.5–10 min 50% A, 10–13.1 min 5% A, 13.1–15 min 5% A, and flow rate 0.40 mL/min. The injection volume was 2 μL, and the auto-sampler temperature was maintained at 4 °C.

The detection was accomplished using a QTrap 6500+ mass spectrometer (Sciex, Concord, ON, Canada) equipped with an IonDrive turbo V electrospray (ESI) source. The quantitative analysis of metabolites was performed in the multiple reaction monitoring mode (MRM) of triple quadrupole mass spectrometry. Typical ion source parameters were ion spray voltage of +5500/−4500 V, curtain gas of 35 psi, temperature of 400 °C, ion source gas 1 at 60 psi, ion source gas 2 at 60 psi, and DP of ± 100 V.

### 4.4. Data Preprocessing and Annotation

SCIEX Analyst Work Station Software (Version 1.6.3, Framingham, MA, USA) was employed for MRM data acquisition and processing. MS (Mass spectrometry) raw data files (.wiff format) were converted to the TXT format using MSconventer. Then, in-house R program and self-compiled database (Shanghai Biotree Biotech Co., Ltd., Shanghai, China) were applied to peak detection and annotation [42,43].

### 4.5. Statistical Analysis

Principal components analysis (PCA) and orthogonal correction partial least squares discriminant analysis (OPLS-DA) were conducted using SIMCA software (V16.0.2, Sartorius Stedim Data Analytics AB, Umea, Sweden) [44] according to the method described by Bai et al. [45]. The *p*-value of the Student’s *t*-test was less than 0.05, and the Variable Importance in the Projection (VIP) of the OPLS-DA model was greater than 1 to identify the metabolites expressed differently. The derived significant metabolites in the stigmas and petals were used for plotting hierarchical clustering based on the Euclidean distance method, complete linkage, and drawn heat maps using the Pheatmap package in R studio [46]. Volcano plots were used to filter the metabolites of interest based on the Log2 (fold change) and −Log10 (*p* value).

### 4.6. KEGG Function Annotation

The differential metabolites were annotated by the Kyoto Encyclopedia of Genes and Genomes (KEGG) database and mapped to KEGG pathways [47]. Metabolite-set enrichment analysis (MSEA) and metabolic network were carried out using MetaboAnalyst 3.0 [48].

## 5. Conclusions

*Crocus sativus* is an important plant that has been used as food flavoring and traditional medicine for thousands of years, and its worldwide production is increasingly growing. Due to the lack of comprehensive understanding of the metabolite composition, the petals constitute a large quantity of by-products that have no industrial application. To the best of our knowledge, the difference in components in the saffron stigmas and petals have not been previously investigated.

In this study, the components of the stigmas and petals in *C. sativus* have been analyzed by means of widely targeted metabolomics. Over 800 metabolites were detected and grouped into 35 classes, and the metabolite composition of the stigmas and petals were generally consistent. These results provide insight into the metabolite composition of the stigmas and petals and may be helpful in understanding and explaining the pharmacological function of saffron extract.

Although the relative content of the 339 metabolites was significantly different, of which 55 were up-regulated and 284 were down-regulated in the petals, the contents of most flavonoids, alkaloids, and coumarins in the petals were not significantly different from those in the stigmas. Furthermore, the petals are rich in monoterpenoids, diterpenes, amino acids, carbohydrates, and vitamins, indicating that they can be used as raw materials for spices or functional drinks. By optimizing the extraction method of flavonoids, alkaloids, and coumarins in the petals, petal extracts can be applied to medicine, food, and cosmetics industries, providing high economic value. It will not only make the full use of the petals but also improve the output value of *C. sativus* cultivation. 

## Figures and Tables

**Figure 1 plants-11-02427-f001:**
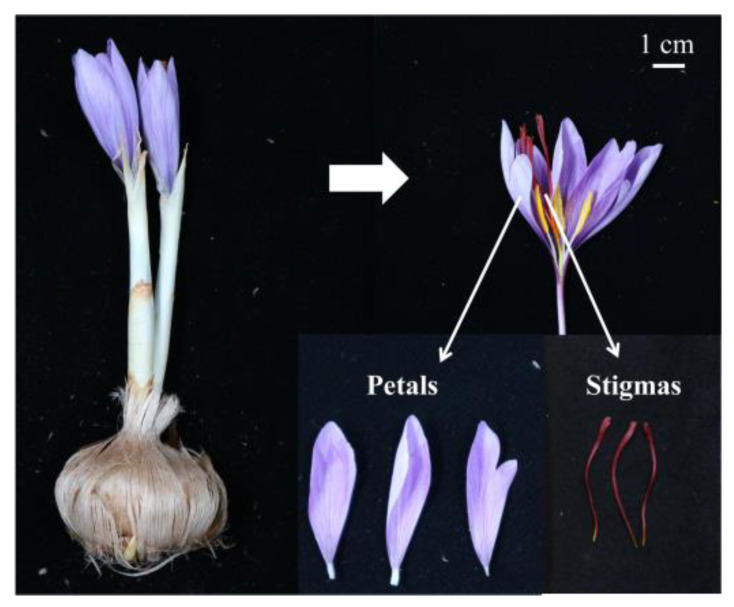
The stigmas and petals of saffron (*Crocus sativus* L.).

**Figure 2 plants-11-02427-f002:**
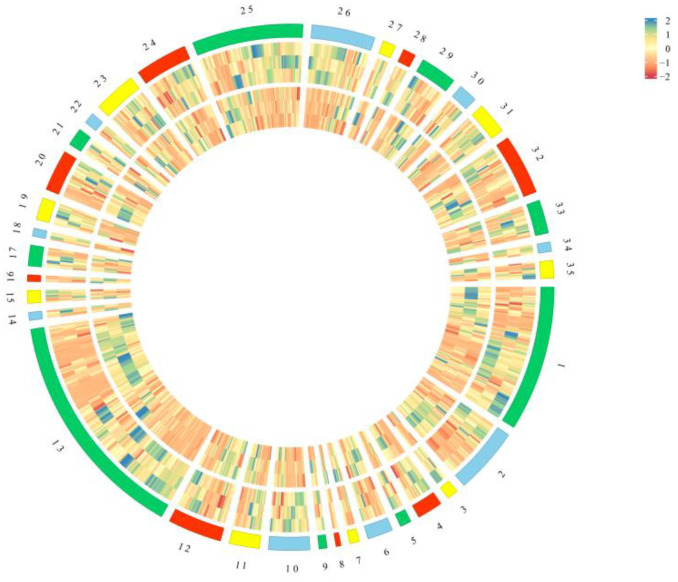
The stigmas and petals of saffron (*Crocus sativus* L.). Circular diagram of all metabolites in the stigmas and petals. All metabolites are classified into 35 categories based on a self-compiled database (Shanghai Biotree Biotech Co., Ltd., Shanghai, China): 1. alkaloids; 2. amino acids and derivatives; 3. anthraquinones; 4. benzene and substituted derivatives; 5. carbohydrates; 6. carboxylic acids and derivatives; 7. chalcones; 8. cholines; 9. cinnamic acids and derivatives; 10. coumarins; 11. diterpenoids; 12. fatty acyls; 13. flavonoids; 14. indoles and derivatives; 15. iridoids; 16. keto acids and derivatives; 17. lignans; 18. lipids; 19. monoterpenoids; 20. nucleotide and its derivates; 21. organic acids; 22. organonitrogen compounds; 23. organooxygen compounds; 24. others (alcohols, glycerolipids, polyols, lactones, et al.); 25. phenols and phenol esters; 26. phenylpropanoids; 27. phytohormone; 28. prenol lipids; 29. pteridines and derivatives, purine nucleosides, pyridines and derivatives, pyrimidine nucleosides, pyrimidine nucleotides, and pyrrolopyrimidines; 30. quinones; 31. sesquiterpenoids; 32. steroids and steroid derivatives; 33. triterpenoids; 34. vitamins; and 35. xanthones.

**Figure 3 plants-11-02427-f003:**
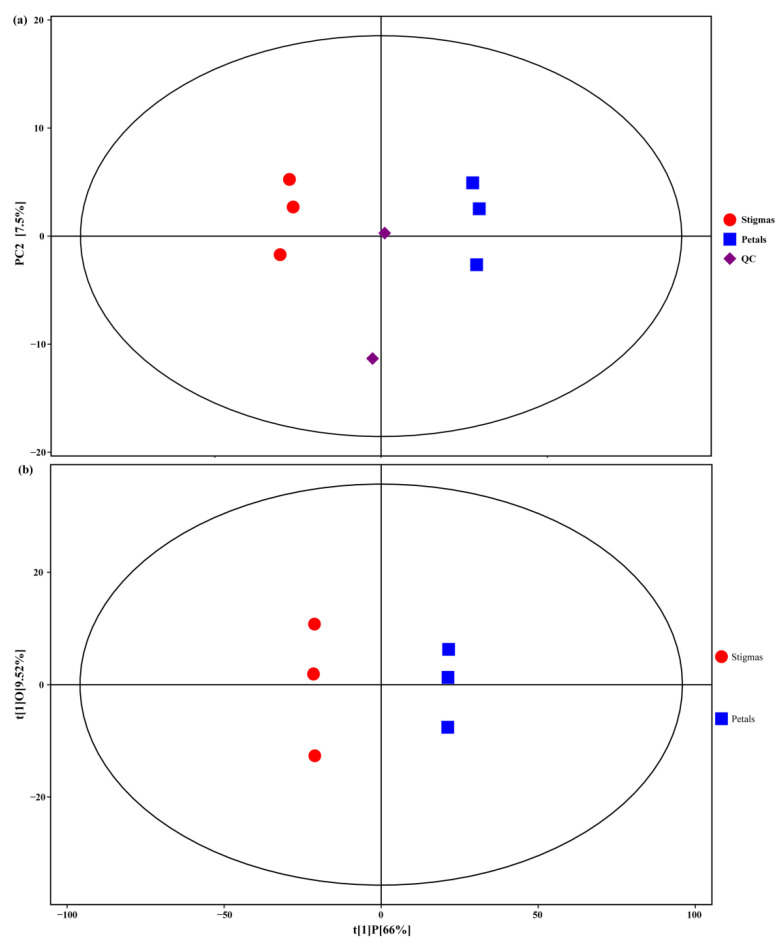
(**a**) PCA score plot of the metabolites in the stigmas and petals. (**b**) Score scatter plot of the OPLS-DA model for the stigmas vs. petals.

**Figure 4 plants-11-02427-f004:**
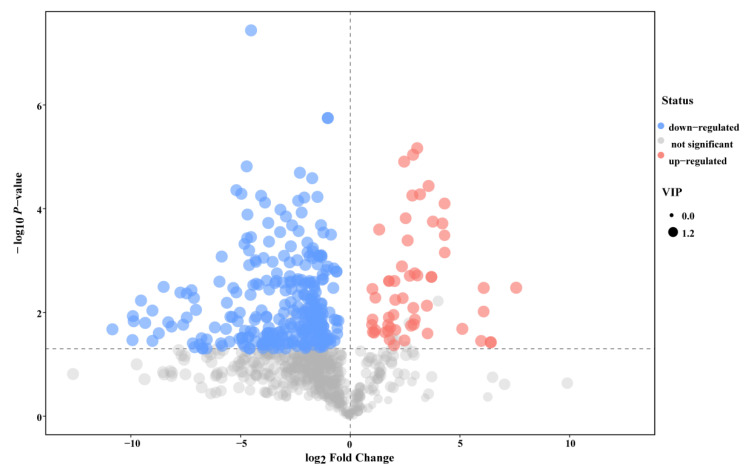
Volcano plot of differential metabolites in the stigmas and petals. Up- and down-regulated metabolites are indicated in red and blue, respectively. Gray represents metabolites that have no significant change.

**Figure 5 plants-11-02427-f005:**
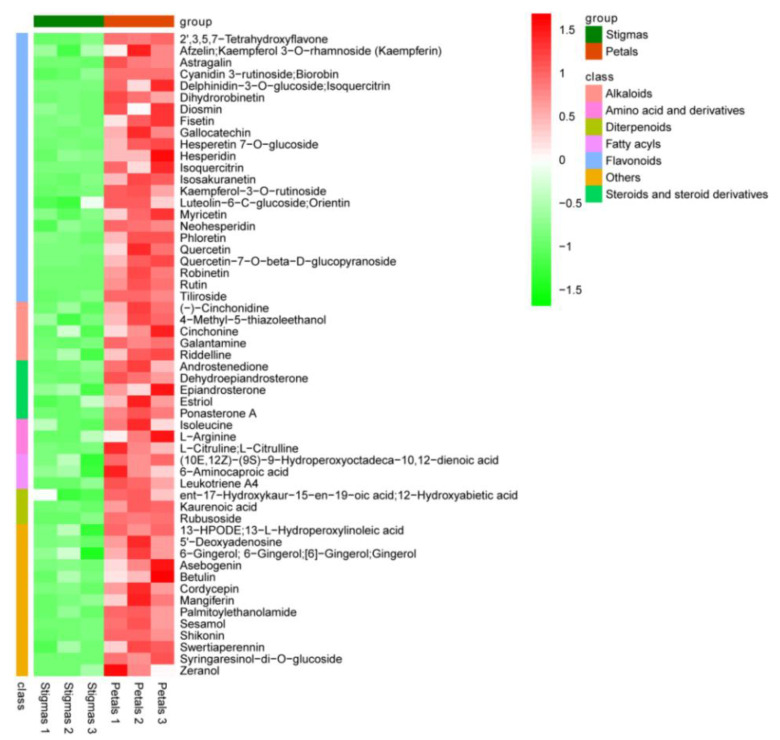
Heat map of the top 55 up-regulated metabolites. The abscissa in the figure represents different experimental samples, and the ordinate represents the differential metabolites compared in the group. Red or green in heatmap indicates high or low expression, respectively, according to the color bar in logarithmic scale shown above the heatmap. The same setup applies to the following Figure 6.

**Figure 6 plants-11-02427-f006:**
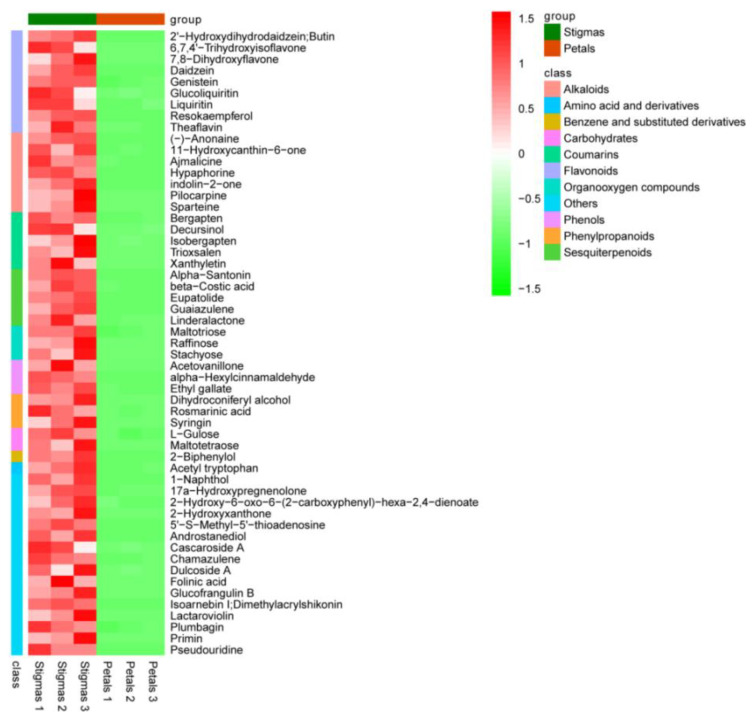
Heat map of the top 55 down-regulated metabolites.

**Figure 7 plants-11-02427-f007:**
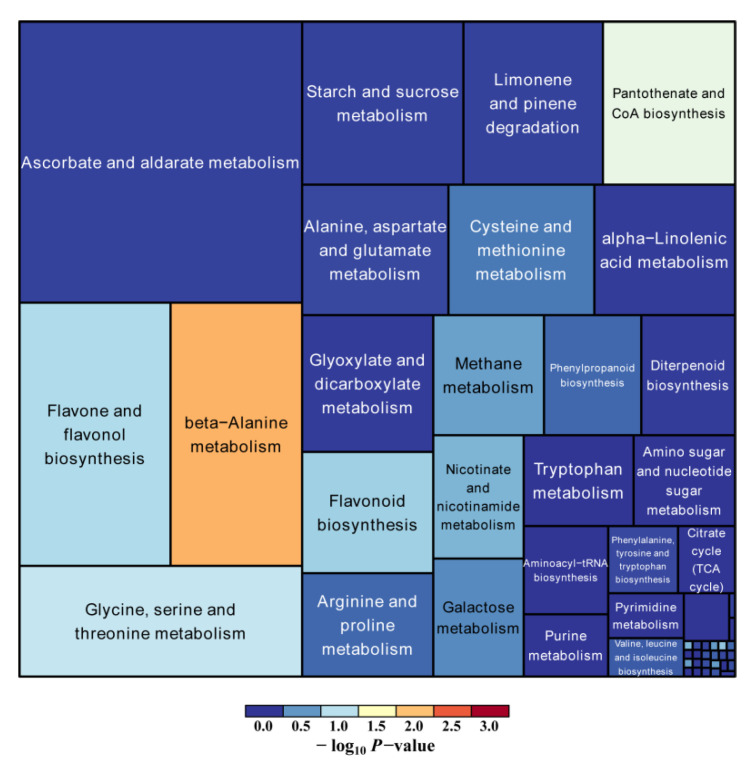
Heat map of the top 55 down-regulated metabolites. Pathway analysis for petals vs. stigmas. One metabolic pathway was represented by one rectangular in the diagram. The sizes indicate the influencing factor of the pathway in the topology analysis, with a larger size showing a larger degree of the influencing factor. The colors represent the *p* values (negative natural logarithm, −ln(*p*)) of the enrichment analysis, with darker colors showing a higher degree of enrichment.

**Figure 8 plants-11-02427-f008:**
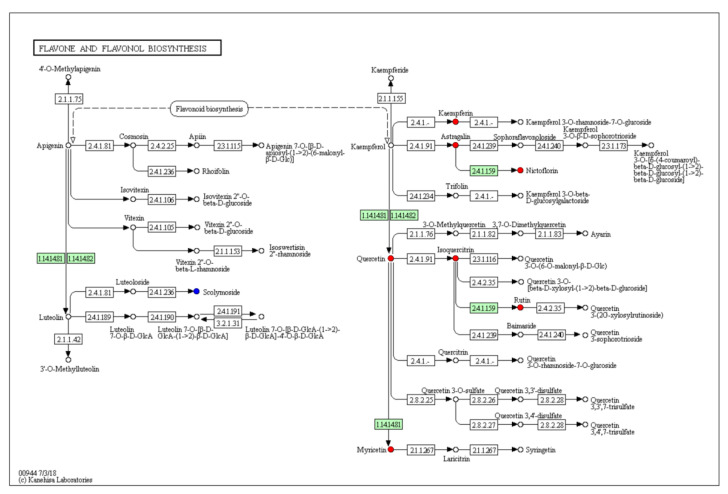
Flavone and flavonol biosynthesis (https://www.kegg.jp/entry/map00944 (accessed on 10 January 2022)). Blue and red round dots represent down-regulated metabolite and up-regulated metabolite, respectively.

**Figure 9 plants-11-02427-f009:**
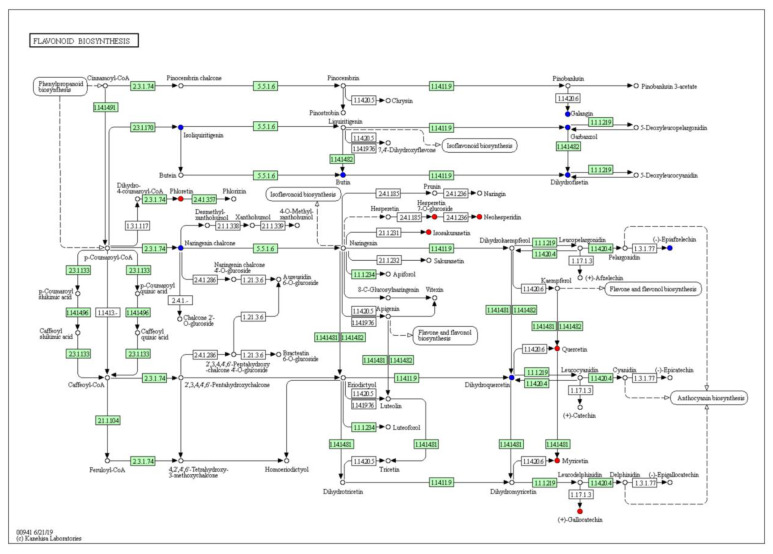
Flavonoid biosynthesis (https://www.kegg.jp/entry/map00941 (accessed on 10 January 2022)). Blue and red round dots represent down-regulated metabolite and up-regulated metabolite, respectively.

## Data Availability

The data that support the findings of this study are available from the corresponding authors upon reasonable request.

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
