# Peer review of "Comparative Metabolomics Analysis of Stigmas and Petals in Chinese Saffron (Crocus sativus) by Widely Targeted Metabolomics"

_plants, 2022, doi:10.3390/plants11182427_

Round 1

Reviewer 1 Report

The authors report an interesting analysis on the secondary metabolites of the reproductive and vexillary organs of the Crocus sativus flower. In particular, the authors identified about 900 metabolites clustered in alkaloids, amino acids, anthraquinones, flavonoids, phenols, terpenoids etc.

Although the study applies innovative analytical technologies, the experimental design is very obvious. Indeed, this is not the first work that analyzes the content of secondary metabolites in the flower of Crocus sativus. Authors should better discuss the innovativeness and limitations of the manuscript.

Furthermore, the authors report that some metabolites are up-regulated or down-regulated. What are they referring to? Did they perform a treatment? What is the decrease or increase due to?

Author Response

Dear reviewer,

Thank you very much for your advice on our manuscript. We appreciate the time and effort that you dedicated to providing feedback on our manuscript and are grateful for the insightful comments on and valuable improvements to our manuscript.

Answers to reviewers:

  • Although the study applies innovative analytical technologies, the experimental design is very obvious. Indeed, this is not the first work that analyzes the content of secondary metabolites in the flower of Crocus sativus. Authors should better discuss the innovativeness and limitations of the manuscript.

Reply: Thank you for your comments. 

In the Introduction, we supplement the current research progress of stigmas and petals composition analysis (Line 63-70).

We have expanded the Discussion to add the advantages of non target metabolomics analysis in metabolite analysis, and the application prospects of stigma and petal metabolite analysis (Line 206-214, 257-268).

Furthermore, the authors report that some metabolites are up-regulated or down-regulated. What are they referring to? Did they perform a treatment? What is the decrease or increase due to?

Reply: Thank you for your comments. 

The samples used for testing were not specially treated. In our manuscript, the 'up regulation' or 'down regulation' is used to indicate the high or low relative content of the metabolite in the petals

The description methods simply refer to the reported literatures.

Dossou SSK, Xu F, Cui X, Sheng C, Zhou R, You J, Tozo K, Wang L. Comparative metabolomics analysis of different sesame (Sesamum indicum L.) tissues reveals a tissue-specific accumulation of metabolites. BMC Plant Biol, 2021, 21(1):352.

Shao, F., Zhang, L., Guo, J. et al. A comparative metabolomics analysis of the components of heartwood and sapwood in Taxus chinensis (Pilger) Rehd. Sci Rep, 2019, 9: 17647.

Liu P, Weng R, Xu Y, Feng Y, He L, Qian Y, Qiu J. Metabolic Changes in Different Tissues of Garlic Plant during Growth. J Agric Food Chem, 2020, 68(44): 12467-12475.

Best regards,

Lin Zhou

Reviewer 2 Report

Good work, with possible interesting applications in agronomy and food chemistry. Good application of chemometrics. Only few trivial typos must be corrected: see list below.

Line 11: ERRATA: “are” CORRIGE: “is”

Lines 17 and 23 and 84 and 91 and 103 and 139: ERRATA: “amino acid” CORRIGE: “amino acids”

Lines 19 and 307: ERRATA: “component” CORRIGE: “components”

Line 29: ERRATA: “has” CORRIGE: “have”

Line 37: ERRATA: “is a widely” CORRIGE: “is widely”

Line 39: ERRATA: “Although, China” CORRIGE: “Although China”

Line 52: ERRATA: “applications” CORRIGE: “application”

Line 65: ERRATA: “technologies” CORRIGE: “technology”

Line 67: ERRATA: “an” CORRIGE: “a”

Line 108: ERRATA: “seven” CORRIGE: “7”

Lines 109 and 110 : ERRATA: “eight” CORRIGE: “8”

Line 113: ERRATA: “six” CORRIGE: “6”

Line 122: ERRATA: “indicating that the” CORRIGE: “indicating the”

Line 151: keep the caption in teh same page as the figure

Line 154: ERRATA: “diferential” CORRIGE: “differential”

Line 182: ERRATA: “which produced” CORRIGE: “which is produced”

Line 204: ERRATA: “its” CORRIGE: “their”

Line 324: ERRATA: “as” CORRIGE: “are”

Line 337: ERRATA: “it” CORRIGE: “they”

Author Response

Dear reviewer,

Thank you very much for your advice on our manuscript. We appreciate the time and effort that you dedicated to providing feedback on our manuscript and are grateful for the insightful comments on and valuable improvements to our manuscript.

We have incorporated most of the suggestions. Those changes are highlighted within the manuscript. Here below is our description on revision according to the comments.

Line 11: ERRATA: “are” CORRIGE: “is”

Reply: Thank you for your comments. 

We changed the sentence to ‘The dried stigmas of Crocus sativus, commonly known as saffron, are consumed largely worldwide because of its high edible values and beneficial biological activities for health.

Lines 17 and 23 and 84 and 91 and 103 and 139: ERRATA: “amino acid” CORRIGE: “amino acids”

Lines 19 and 307: ERRATA: “component” CORRIGE: “components”

Line 29: ERRATA: “has” CORRIGE: “have”

Line 37: ERRATA: “is a widely” CORRIGE: “is widely”

Line 39: ERRATA: “Although, China” CORRIGE: “Although China”

Line 52: ERRATA: “applications” CORRIGE: “application”

Line 65: ERRATA: “technologies” CORRIGE: “technology”

Line 67: ERRATA: “an” CORRIGE: “a”

Line 122: ERRATA: “indicating that the” CORRIGE: “indicating the”

Line 151: keep the caption in teh same page as the figure

Line 154: ERRATA: “diferential” CORRIGE: “differential”

Line 182: ERRATA: “which produced” CORRIGE: “which is produced”

Line 204: ERRATA: “its” CORRIGE: “their”

Line 324: ERRATA: “as” CORRIGE: “are”

Line 337: ERRATA: “it” CORRIGE: “they”

Reply: Thank you for your comments. The above words had been modified, and the modified content is highlighted in red.

Line 108: ERRATA: “seven” CORRIGE: “7”

Lines 109 and 110 : ERRATA: “eight” CORRIGE: “8”

Line 113: ERRATA: “six” CORRIGE: “6”

Reply: Thank you for your comments. We refer to the writing norms and requirements of most journals. Specifically, numbers one to nine without units should be presented as words, and numbers 10 and over without units should be presented as numerals. Thus, numbers under 10 without units were presented as words in our manuscript.

Thank you and best regards.

Lin Zhou

Reviewer 3 Report

This manuscript deals with an untargeted metabolomics approach to unravel the metabolic content and profile of two different parts of the crocus sativus plant, petals and stigmas. The latter is widely known as saffron and so far, there's plenty of literature findings about its chemical composition and how it can vary with different pre- and post-harvest processes. Much less is known about petals although there are some very insightful recent publications on this topic e.g.

Mottaghipisheh, J., Mahmoodi Sourestani, M., Kiss, T., Horváth, A., Tóth, B., Ayanmanesh, M., Khamushi, A., & Csupor, D. (2020). Journal of pharmaceutical and biomedical analysis184, 113183.

Xu, S., Ge, X., Li, S., Guo, X., Dai, D., & Yang, T. (2019). Chemistry & biodiversity16(10), e1900363. 

(not cited in the manuscript)

The authors used their experimentaly grown plants to track the metabolic "signature" of petals and compare with that of the commercially important stigmas but their experimental design presents some gaps which are not clearly addressed. For example, they do not argue on the representativeness of the samples, they used only one sample per plant part (I guess three replicates, it is not clear) with no reference (e.g. conventionally grown or dried stigmas) and applied an advanced supervised discriminant technique (OPLS-DA) to show that the composition of those samples is different. Can these results be reproduced with plant material from another harvest season or another location? There was no discussion about previously published results on chemical variability. And poor discussion about the volatiles, as well.

The hints about up- and down-regulated metabolites are very useful for unraveling metabolic pathways but still, the methods used to identify peaks and annotate them are far from being transparent as in-house built algorithms and private databases were used. So how can anyone evaluate the accuracy of the chromatographic/spectroscopic results? 

I think that the title, abstract and content especially in the experimental and the Disucssion sections have to be substantially revised/enriched with new data to support and reflect what was really measured in this study and how the results help to advance the state-of-the-art knowledge about saffron's plant biochemistry 

Author Response

Dear reviewer,

Thank you very much for your advice on our manuscript. We appreciate the time and effort that you dedicated to providing feedback on our manuscript and are grateful for the insightful comments on and valuable improvements to our manuscript.

Answers to reviewers:

  • This manuscript deals with an untargeted metabolomics approach to unravel the metabolic content and profile of two different parts of the crocus sativus plant, petals and stigmas. The latter is widely known as saffron and so far, there's plenty of literature findings about its chemical composition and how it can vary with different pre- and post-harvest processes. Much less is known about petals although there are some very insightful recent publications on this topic e.g.

Mottaghipisheh, J., Mahmoodi Sourestani, M., Kiss, T., Horváth, A., Tóth, B., Ayanmanesh, M., Khamushi, A., & Csupor, D. (2020). Journal of pharmaceutical and biomedical analysis184, 113183.

& Xu, S., Ge, X., Li, S., Guo, X., Dai, D., & Yang, T. (2019). Chemistry & biodiversity16(10), e1900363. 

(not cited in the manuscript)

Reply: Thank you for your comments. 

We have added these related research progress in the Introduction section

  • The authors used their experimentaly grown plants to track the metabolic "signature" of petals and compare with that of the commercially important stigmas but their experimental design presents some gaps which are not clearly addressed. For example, they do not argue on the representativeness of the samples, they used only one sample per plant part (I guess three replicates, it is not clear) with no reference (e.g. conventionally grown or dried stigmas) and applied an advanced supervised discriminant technique (OPLS-DA) to show that the composition of those samples is different.

Reply: Thank you for your comments. 

The level and type of secondary metabolites is strongly influenced by the geoclimatic characteristics of the cultivation area as well as the preparation procedures and traditions followed in that area. On the one hand, through PCA and OPLS-DA analysis, we confirmed that the samples had good intra group repeatability and differences between groups. On the other hand, significant differences between metabolites of petals and stigmas control groups were identified using variable importance in projection (VIP) from OPLS-DA .

  • Can these results be reproduced with plant material from another harvest season or another location? There was no discussion about previously published results on chemical variability. And poor discussion about the volatiles, as well.

Reply: Thank you for your comments. 

Currently, the "two-segment" cultivation mode has now been endorsed and scaled up throughout the country in China. Under this cultivation mode, the flowering period of saffron is concentrated in late October to early November. We believe that different cultivation sites or harvesting season may affect the absolute content of stigma and petal metabolites, but have little impact on the composition and relative content of their metabolites.

We have modified the Discussion to better discussion about the volatiles.

  • The hints about up- and down-regulated metabolites are very useful for unraveling metabolic pathways but still, the methods used to identify peaks and annotate them are far from being transparent as in-house built algorithms and private databases were used. So how can anyone evaluate the accuracy of the chromatographic/spectroscopic results? 

Reply: Thank you for your comments. 

The methods used to identify peaks and annotate them are mainly referred to the following literatures.

Chen W, Gong L, Guo Z, Wang W, Zhang H, Liu X, Yu S, Xiong L, Luo J. A novel integrated method for large-scale detection, identification, and quantification of widely targeted metabolites: application in the study of rice metabolomics. Mol Plant, 2013, 6(6): 1769-80.

Shi X, Wang S, Jasbi P, Turner C, Hrovat J, Wei Y, Liu J, Gu H. Database-Assisted Globally Optimized Targeted Mass Spectrometry (dGOT-MS): Broad and Reliable Metabolomics Analysis with Enhanced Identification. Anal Chem, 2019, 91(21): 13737-13745.

Chen Y, Zhou Z, Yang W, Bi N, Xu J, He J, Zhang R, Wang L, Abliz Z. Development of a Data-Independent Targeted Metabolomics Method for Relative Quantification Using Liquid Chromatography Coupled with Tandem Mass Spectrometry. Anal Chem, 2017, 89(13): 6954-6962.

Zheng F, Zhao X, Zeng Z, Wang L, Lv W, Wang Q, Xu G. Development of a plasma pseudotargeted metabolomics method based on ultra-high-performance liquid chromatography-mass spectrometry. Nat Protoc, 2020, 15(8): 2519-2537.

Gu H, Zhang P, Zhu J, Raftery D. Globally Optimized Targeted Mass Spectrometry: Reliable Metabolomics Analysis with Broad Coverage. Anal Chem, 2015, 87(24): 12355-12362.

Luo P, Dai W, Yin P, Zeng Z, Kong H, Zhou L, Wang X, Chen S, Lu X, Xu G. Multiple reaction monitoring-ion pair finder: a systematic approach to transform nontargeted mode to pseudotargeted mode for metabolomics study based on liquid chromatography-mass spectrometry. Anal Chem, 2015, 87(10): 5050-5055.

Luo P, Yin P, Zhang W, Zhou L, Lu X, Lin X, Xu G. Optimization of large-scale pseudotargeted metabolomics method based on liquid chromatography-mass spectrometry. J Chromatogr A,  2016, 1437:127-136.

Chen S, Kong H, Lu X, Li Y, Yin P, Zeng Z, Xu G. Pseudotargeted metabolomics method and its application in serum biomarker discovery for hepatocellular carcinoma based on ultra high-performance liquid chromatography/triple quadrupole mass spectrometry. Anal Chem, 2013, 85(17):8326-8333.

Zha H, Cai Y, Yin Y, Wang Z, Li K, Zhu ZJ. SWATHtoMRM: Development of High-Coverage Targeted Metabolomics Method Using SWATH Technology for Biomarker Discovery. Anal Chem, 2018, 90(6):4062-4070

MRM data processing are mainly referred to the following three literatures.

Smith CA, Want EJ, O'Maille G, Abagyan R, Siuzdak G. XCMS: processing mass spectrometry data for metabolite profiling using nonlinear peak alignment, matching, and identification. Anal Chem, 2006, 78(3): 779-787. 

Kuhl C, Tautenhahn R, Bottcher C, Larson TR, Neumann S. CAMERA: an integrated strategy for compound spectra 438 extraction and annotation of liquid chromatography/mass spectrometry data sets. Analytical Chemistry, 2012, 84 (1): 283-289.

Zhang ZM. Xia T, Peng Y, Pan M, Zhang MJ, Lu HM, Chen XQ, Liang YZ. Multiscale peak detection 440 in wavelet space. Analyst, 2015, 140: 7955-7964.

Although data analysis was performed using a self-compiled database (Shanghai Biotree Biotech Co., Ltd., Shanghai, China), the database has been used for metabolome analysis of multiple species. The relevant research carried out by this method have been published in Food Chemistry, Scientia Horticulturae, Food Bioscience, Foods and other journals.

Relevant literature information is as follows:

Liu Y, Liu J, Wang R, Sun H, Li M, Strappe P, Zhou Z. Analysis of secondary metabolites induced by yellowing process for understanding rice yellowing mechanism. Food Chemistry, 2020, 342: 128204.

Mamat A, Tusong K, Xu J. Identification of metabolic pathways related to rough-skinned fruit formation in korla pear. Scientia Horticulturae, 2021, 288: 110414.

Li C, Xu T, Liu X, Wang X, Xia T. The expression of β-glucosidase during natto fermentation increased the active isoflavone content. Food Bioscience, 2021, 43: 101286.

Fan W, Li B, Tian H, Li X, Ren H, Zhou Q. Metabolome and transcriptome analysis predicts metabolism of violet-red color change in Lilium bulbs. Journal of the Science of Food and Agriculture, 2021, 102: 2903-2915.

Zou C, Lu T, Wang R, Xu P, Jing Y, Wang R, Xu J, Wan J. Comparative physiological and metabolomic analyses reveal that Fe3O4 and ZnO nanoparticles alleviate Cd toxicity in tobacco. Journal of Nanobiotechnology, 2022, 20: 1-22.

Ma Y, Li J, Li J, Yang L, Wu G, Liu S. Comparative Metabolomics Study of Chaenomeles speciosa (Sweet) Nakai from Different Geographical Regions. Foods, 2022, 11: 1019.

Zhu L, Tian Y, Ling J, Gong X, Sun J, Tong L. Effects of Storage Temperature on Indica-Japonica Hybrid Rice Metabolites, Analyzed Using Liquid Chromatography and Mass Spectrometry. International Journal of Molecular Sciences, 2022, 23: 7421.

Therefore, we believe that the method used in this experiment is feasible and the data obtained are reliable.

  • 85. What kind of phenol esters? Why is this distinction made with esters and not with acids?Please define which database was used for compound classification.

Reply: Thank you for your comments. 

We detected three phenolic esters, including phenyl acetate, trimethoprim, and venlafaxine.

For the chemical classification of metabolites, information was retrieved from the database PubChem  and HMDB.

  • 211-217. basic metabolites were not found annotated in the heat maps. How were they classified? Do you think that you would have the same results if you analysed conventional saffron?

Reply: Thank you for your comments. 

The classification of monoterpenes and diterpenoids is shown in Fig. 2. Our results showed that the relative contents of these substances in stigma and petals were not significantly different. Since these metabolites are not significantly up-regulated or down-regulated in petals compared with stigmas, they are not shown in Fig 5 and Fig 6. 

“Do you think that you would have the same results if you analysed conventional saffron?”——I agree that it needs to be taken into consideration. We do not choose traditional saffron as a reference, mainly for the following reasons.

Freeze-drying has been reported to maintain the appearance, shape, flavor, and biological activities of food items, making it a promising drying technique. Therefore, most studies of metabolome analysis used freeze-dried test samples. At present, most of the Chinese saffron production enterprises use microwave drying method to process the stigmas. As per our earlier findings, the water content of petals was significantly higher than that of stigmas. For the same weight of tissue, petals need longer drying time than stigmas. Considering the effect of high temperature on the metabolites in the test samples, we used freeze-drying method to better understand the metabolite composition of stigma and petal. 

  • I think that the title, abstract and content especially in the experimental and the Disucssion sections have to be substantially revised/enriched with new data to support and reflect what was really measured in this study and how the results help to advance the state-of-the-art knowledge about saffron's plant biochemistry.

Reply: Thank you for your comments. 

Can we add supplementary materials (Compound names, retention times, exact mass, and KEGG IDs of detected compounds) to support our results? Can these additions resolve this issue? If feasible, we will upload the relevant data.

Thank you and best regards.

Lin Zhou

Reviewer 4 Report

In the present manuscript authors described results of their research on metabolite patterns in petals and stigmas of Crocus sativus. The manuscript is written well. The experimental design meets modern research requirements. Results obtained are described in enough details. Discussion made is logical. As whole the manuscript presents valuable information that would be fundamental base for development ecologically friendly approaches for utilization of by-products of this widely used plant species. May be, the manuscript could be improved if the authors enrich information in the conclusion about their point of view for further prospects for using by-products. For example what kind of functional drinks could be prepared? How can be solved the problem with alkaloids content when this material will be used in food and cosmetic systems?

Author Response

Dear reviewer,

Thank you very much for your advice on our manuscript. We appreciate the time and effort that you dedicated to providing feedback on our manuscript and are grateful for the insightful comments on and valuable improvements to our manuscript.

Answers to reviewers:

  • May be, the manuscript could be improved if the authors enrich information in the conclusion about their point of view for further prospects for using by-products. For example what kind of functional drinks could be prepared?

Reply: Thank you for your comments. 

In the Discussion, we have added the development prospect of functional drinks (Line 257-268).

  • How can be solved the problem with alkaloids content when this material will be used in food and cosmetic systems?

Reply: Thank you for your comments. 

The main components of saffron are crocetin and its glucosidic derivatives such as crocin, picrocrocin, safranal, and flavonoids. Thus, past studies have focused more on four main bioactive constitutes.

In recent years, the content detection and extraction of alkaloids, flavonoid and other components have gradually attracted the attention of researchers. The concentrations of alkaloids and flavonoid in the water extract of the dried flowers were 2.4 mg/g and 11.2 mg/g, respectively (Mir et al, 2016). We detected a total of 92 alkaloids in stigmas and petals, including caffeine, theobromine, theophylline, theophylline, galantamine, etc. We will determine the content of different alkaloids in the follow-up study.

In addition, we speculate that the alkaloids in saffron play an active role. On one hand, saffron has a long history of use as traditional medicine to treat diseases in China. Meanwhile, saffron is a precious spice used worldwide as food additive for its coloring and flavoring properties. On the other hand, some researchers believe that effects of saffron extracts on neuropathic pain could be due to their flavonoids, tannins, anthocyanins, alkaloids, and saponins (Safakhah et al, 2016). Furthermore, the potential activity of saffron and its ingredients against cancer has been investigated, and the results show saffron’s potent anticancer activity in preclinical settings, importantly without adverse effects on normal cells (Lambrianidou et al, 2021).

Best regards,

Lin Zhou 

Round 2

Reviewer 1 Report

The authors improved their manuscript following the recommendations of the Reviewers. Now the manuscript can be published.

Author Response

thanks